# Bionomics and population dynamics of anopheline larvae from an area dominated by fish farming tanks in northern Brazilian Amazon

**Ledayane Mayana Costa Barbosa**[1]**, Vera Margarete Scarpassa**[2]*

**1** Departamento de Ciências Biológicas e da Saúde, Laboratório de Arthropoda, Universidade Federal do Amapá, Rodovia Juscelino Kubitschek, Macapá, Amapá, Brasil, **2** Laboratório de Genética de Populações e Evolução de Mosquitos Vetores de Malária e Dengue, Instituto Nacional de Pesquisas da Amazônia, Coordenação de Biodiversidade, Manaus, Amazonas, Brasil

* vera@inpa.gov.br

**Data Availability Statement:** All relevant data are within the manuscript and its Supporting Information files.

## Abstract

### Background

In Brazilian Amazon, deforestation and other anthropogenic activities as a consequence of human occupation have created new and artificial larval habitats for anopheline mosquitoes, providing conditions for oviposition, development and expansion of malaria vector populations.

### Objectives

This study aimed to structurally characterize and describe the entomological and limnological parameters of *Anopheles* larval habitats from a malaria region in northern Brazilian Amazon.

### Methods

Fifty-two larval habitats were investigated in the District of Ilha de Santana, in the Brazilian state of Amapá, comprising fish farming tanks, ponds, and streams. For entomological parameters, the immature larvae were collected monthly from July 2019 to June 2020. For limnological parameters, the water samples were collected from 20 larval habitats during the sampling period. The data were analyzed using Generalized Linear Models, Multivariate analyses, and Kruskal-Wallis tests.

### Results

Fifty habitats were positive for *Anopheles* larvae and a total of nine species were collected. The fish farming tanks had the highest abundance of larvae compared with ponds and streams. *Anopheles darlingi*, *Anopheles nuneztovari* s.l. and *Anopheles triannulatus* were collected in 94% of the larval habitats and showed the highest positivity index. The degree of shade and the type of water of the breeding sites were important factors for the presence

**Funding:** This research was funded by Ministério da Ciência, Tecnologia e Inovação/ Instituto Nacional de Pesquisas da Amazônia, Brazil, (Number: 12.311) to Dr. Vera Margarete Scarpassa. The funders had no role in study design, data collection and analysis, decision to publish, or preparation of the manuscript.

**Competing interests:** The authors have declared that no competing interests exist.

of the main malaria vector, *A. darlingi*. This species was the most affected by pH, total dissolved solids, electrical conductivity, and nitrate.

## Conclusions

Our findings indicate that fish farming tanks are major contributors to vector anopheline abundance and malaria transmission. Vector control strategies focused on these habitats are urgently needed.

## Introduction

In Brazil, ~ 99.7% of human malaria cases occur in the Amazon rainforests, where the transmission occurs mainly by *Plasmodium vivax* (>90%) [1]. The complex environment of the Amazon rainforests favors the development of mosquito populations, which is directly associated with the annual flood cycle and river flow regimes [1,2]. This pattern allied to the tropical climate of the region, among other factors, favors the high density of anopheline mosquitoes [1–3].

In Amapá, one of the Brazilian states that is covered with Amazonian rainforests, the municipalities of Mazagão, Porto Grande, and Santana form together the main endemic corridor of malaria cases, which is primarily explained by the migratory flow of humans. From January to June 2019, these municipalities contributed to approximately 54% of the malaria cases in the state and approximately 43% in the same period of 2020 [4]. This scenario can be attributed to the high density of the main malaria vector, *Anopheles darlingi* [2,5–8]. In addition, entomological investigations conducted in Amapá have reported the association of *Anopheles marajoara* and *Anopheles nuneztovari* s.l. with malaria transmission [2,7–9], contributing to the complexity of the malaria transmission patterns in the region. Further investigations on the vector composition in Amapá can help implement appropriate methods of intervention.

Female *Anopheles* mosquitoes require accumulated water to lay their eggs and allow larval development. These habitats can be natural or artificial water bodies, such as riverbanks, lagoons, rivers, and marshes, preferably containing clean water with organic matter, aquatic vegetation and shading [10,11]. The physical and chemical parameters of aquatic sites have a direct influence on the metabolism and development of anopheline larvae, as they promote the proliferation of algae and other microorganisms that are an essential part of their diet [12,13], and can also influence the distribution and abundance of several anopheline species [14]. Identifying and monitoring habitats used by anopheline mosquito larvae are particularly important, because nutrient availability strongly influences the fitness of mosquito adults, and consequently their longevity, impacting the malaria transmission dynamics.

In the past few decades, deforestation and other anthropogenic activities, followed by human occupation across Amazonia, have created new and artificial habitats for the development of mosquito larvae, such as dams, fish farming tanks, and ponds [13,15,16]. These human-generated habitats provide the ideal conditions for the oviposition and development of the anopheline immatures, particularly *A. darlingi*. In several municipalities of Brazilian Amazon, the local government programs have encouraged and provided resources to residents for implementing fish farming activities. Fish farming tanks are usually installed near human settlements, creating excellent conditions for the anopheline vector to feed on humans [17], and the transmission of the malaria parasite. A recent study conducted in the Municipality of Mâncio Lima, western Acre, Brazil, an area with a high malaria incidence, identified fish farming

tanks as the major contributors to vector abundance and malaria transmission [16]. In this context, identifying the preferred anopheline larval habitats is essential for vector surveillance and providing support for the development of integrated vector control strategies. This study aimed to structurally characterize and investigate the entomological and limnological parameters of the habitats used by *Anopheles* larvae in an area of malaria transmission dominated by fish farming tanks in northern Brazilian Amazon. The results presented here are part of a wider study of adult anophelines and *Plasmodium* transmission in the District of Ilha de Santana, Amapá [18]. This is the first study to investigate these parameters in the Brazilian state of Amapá, and are valuable for the development of integrated vector control strategies.

## Materials and methods

Samples of larvae and the water from breeding sites were collected in the District of Ilha de Santana (00˚04'00''S and 00˚06'00''S; 51˚08'00''W and 51˚12'30''W), located in the Municipality of Santana, Amapá, Brazil (Fig 1) [19]. Ilha de Santana is located in southern Amapá, 26 km from the capital Macapá and comprises an area of 20.06 km$^2$, with an estimated population of 3,226 inhabitants [20]. This district is located on the banks of the Amazon River and is subdivided in urban, transitional, and rural areas. The main economic activities include agriculture (fruits and vegetables), especially the production of fruit pulps. Two important products are obtained from the açaí palm (*Euterpe oleracea* Mart): fruits and palm hearts.

The selection of the study area took into account malaria cases, high density of *A. darlingi* and other vector [2,7–9,18], and the high migratory flow of humans related to intense agricultural activities.

### Structural characterization of larval habitats

Larval habitats were first located using a hydrographic map of the study area provided by the Secretary of the State of Amapá for Environment. These habitats were subsequently confirmed

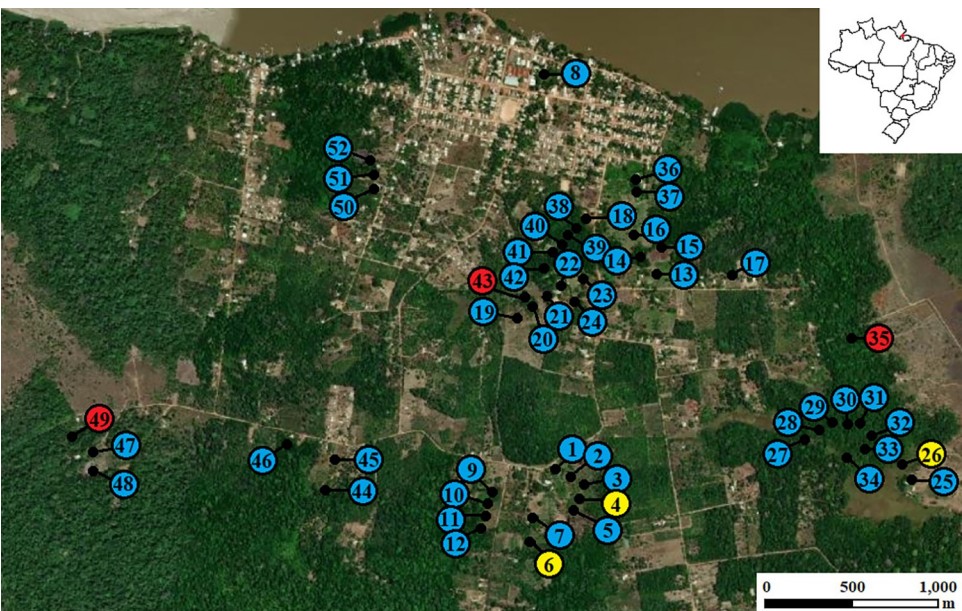

**Fig 1. Satellite image of the District of Ilha de Santana, showing the 52 breeding sites.** Fish farming tanks are represented in blue, ponds in yellow, and streams in red. The small map represents the Brazilian states (Amapá is highlighted in red). The images were generated using https://earthexplorer.usgs.gov/ (public domain) and adapted for graphical purposes.

in the field. The potential breeding sites were numbered, photographed, georeferenced using a Garmin Etrex Legend H GPS, and structurally characterized according to the Bulletin of Immature Collection that provides a detailed classification of larval habitats (Figs 1–4) [21]. The size of the breeding site and the distance from human dwellings were measured. Subsequently, breeding sites were categorized based on the classification of larval habitats (pond, swamp, dam, stream, ditch, canal, fish farming tank, fish pond, among others), shading (none, < 50% or > 50%), water type (clear, muddy, brackish or polluted), presence of debris (trunks/roots, leaves, fruits and/or flowers), current (strong, moderate, weak or none), vegetation (emergent, floating and/or submerged), and the type of the breeding site (permanent, semi-permanent or temporary). Water type was evaluated based on color, but this factor was not associated with the presence of garbage at the breeding sites, as it was not observed at the studied sites.

## Entomological parameters of the immatures

For the analysis of entomological parameters, immatures were collected from all mapped habitats in Ilha de Santana and were classified as permanent or temporary and natural or artificial. A total of 52 larval habitats were identified in the study area, classified as 46 fish farming tanks, three ponds, and three streams (Figs 2–4). All fish farming tanks and ponds mapped in Ilha de Santana were inspected. For the sampling in streams, which flow through extensive areas in the region, two selection criteria were used: 1) Proximity to human settlements and accessibility, since the most remote areas (uninhabited in some areas) had no roads or trails; 2) Conservation state of streams, since waste disposal, including human waste, was observed *in loco* during our survey in some streams, especially those located near the urban area of Ilha de

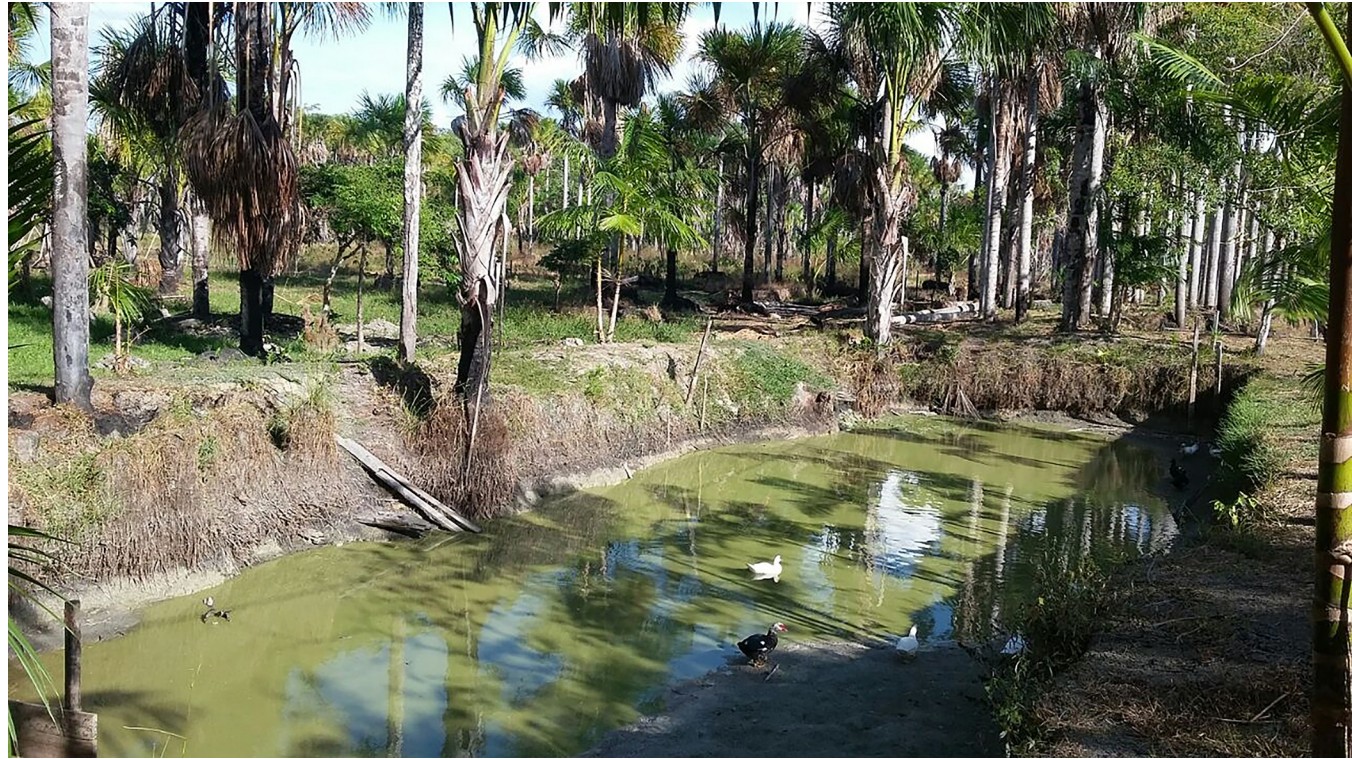

**Fig 2. Image of the fish farming tanks investigated as larval habitats of *Anopheles* in the District of Ilha de Santana, Brazilian state of Amapá.**

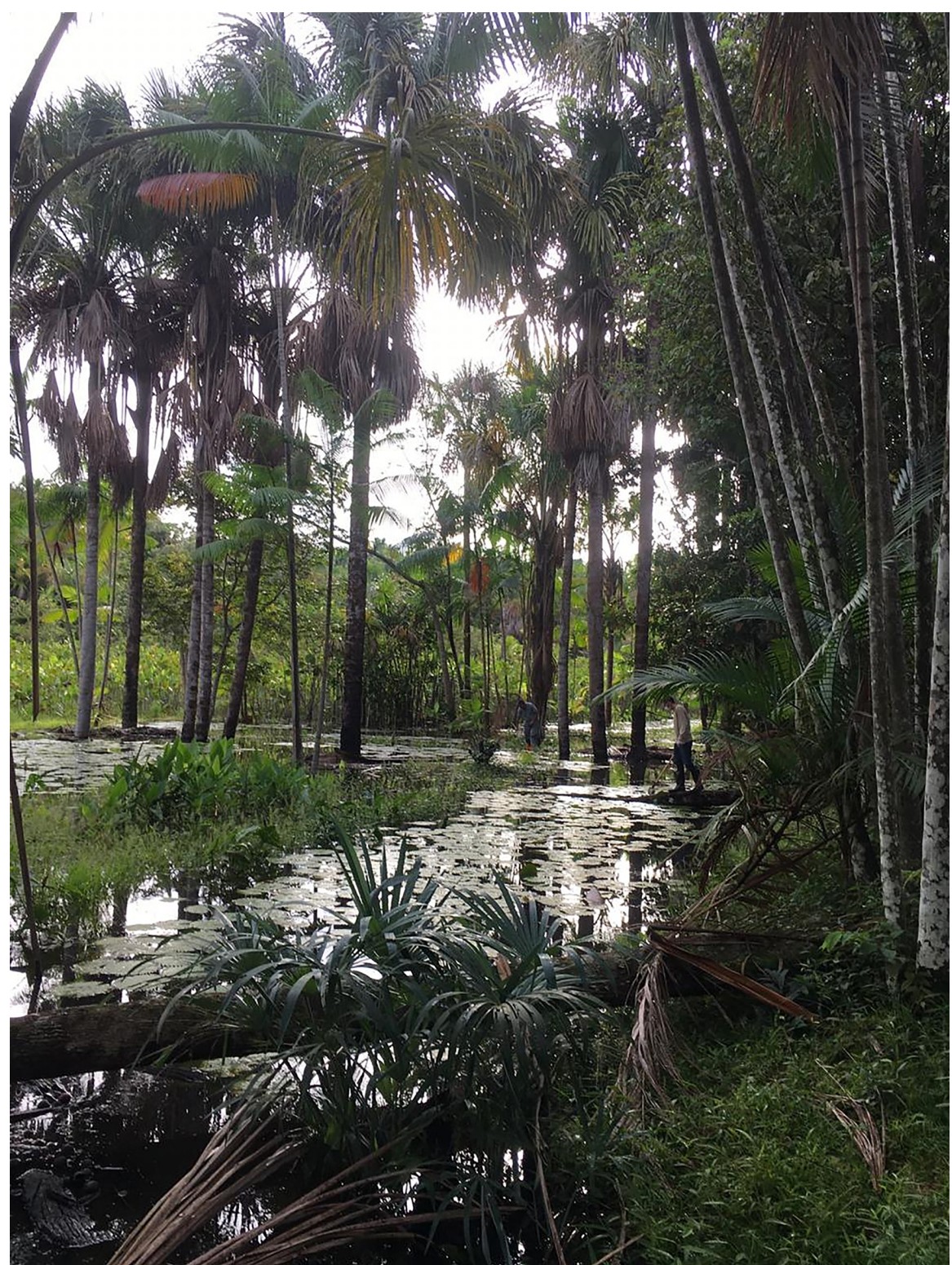

**Fig 3. Image of the ponds investigated as larval habitats of *Anopheles* in the District of Ilha de Santana, Brazilian state of Amapá.**

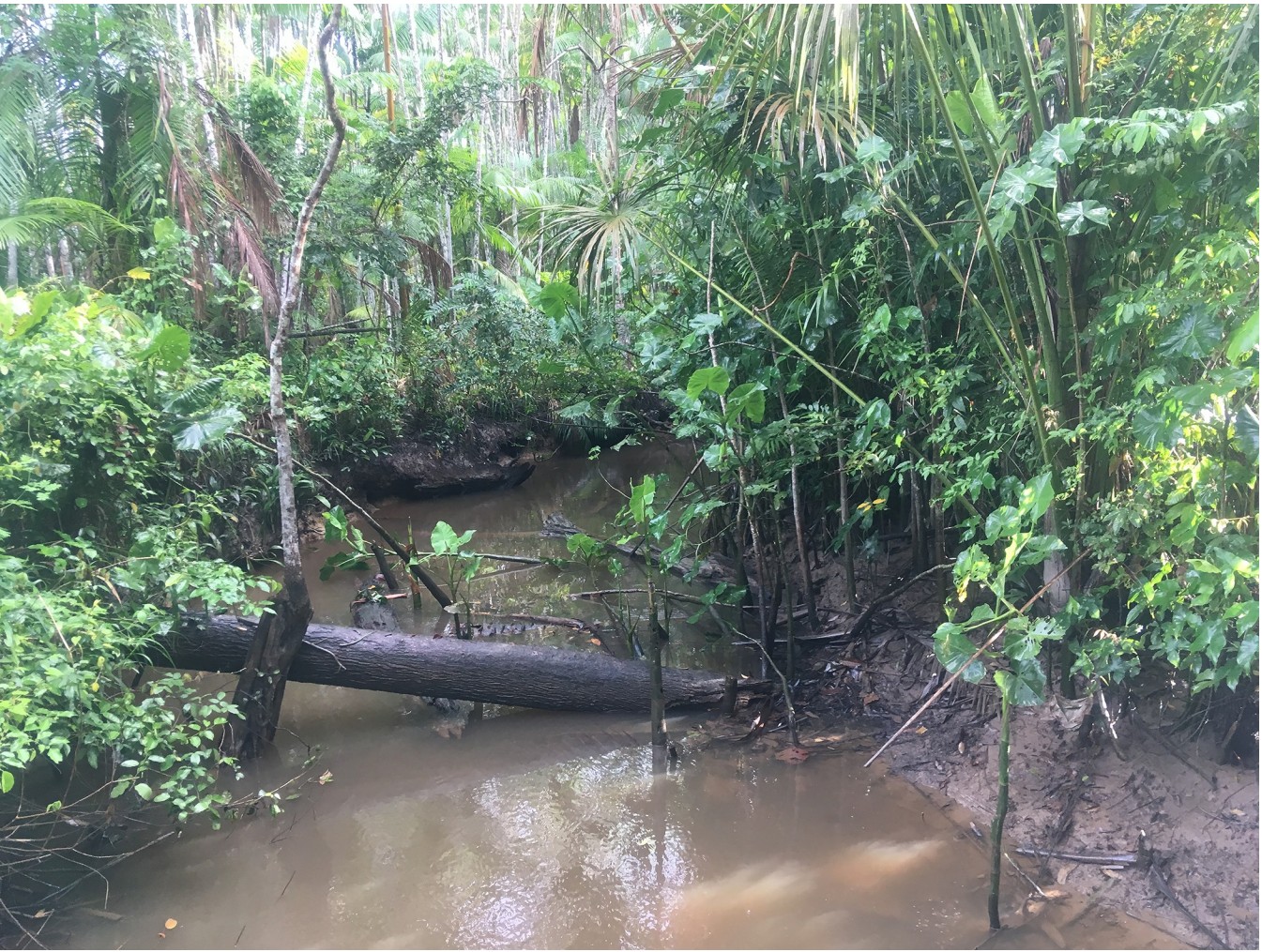

**Fig 4. Image of the streams investigated as larval habitats of *Anopheles* in the District of Ilha de Santana, Brazilian state of Amapá.**

Santana. As anophelines have specific water requirements, streams that did not have these conditions were excluded from the analyses. Thus, only three streams were analyzed.

Monthly collections were carried out from July 2019 to June 2020, totaling a sampling effort of twelve field collections. At the collection in larval habitats was performed in the early morning hours, every 5 m from a defined collection point throughout the entire length of the breeding site. At each point, sampling was performed nine times, three on the right, three on the left, and three ahead, within a radius of 1.0 m from the point determined by the collector, according to guidelines in Technical Note #012—CGPNCM/DIGES/SVS/MS, of 4 June 2007, following the standardized methods used for *Anopheles* larval studies [22]. Immature specimens were collected with the aid of a standard entomological dipper (11 cm in diameter and 350 mL of volume capacity) with a handle 1.0 m long to allow sampling at sites with difficult access [22]. Each time the water was sampled, the volume was recorded and the contents were transferred into a tray. Third and fourth instars larvae were transferred into plastic vials filled with 70% alcohol and later morphologically identified. First- and second-instar larvae were placed in plastic jars with water from the breeding site and reared until they reached the 4th instar to be identified; pupae were kept in plastic vials with water from the breeding site until

the emergence of adults to be identified. All vials were labeled with date and breeding site number. For the species identification, dichotomous keys of Gorham et al. [23] and Faran and Linthicum [24] were used. *Anopheles intermedius* identified in this study is now assigned to *Anopheles medialis* (*Anopheles intermedius* = *Anopheles medialis*) [25]. The molecular identification was based on the DNA barcoding region of the mitochondrial *COI* gene for specimens of the *A. oswaldoi* complex, which confirms the presence of *Anopheles konderi* in the study area.

The entomological parameters evaluated for the descriptive analysis of immatures were [21,26]:

**Larval Index per Man/Hour (LIMH)** to estimate larval density expressed by the equation:

$$LIMH = \sum_{j=1}^{L} \frac{\frac{N}{(Cxh)}}{S}$$ where: $N$ = number of larvae; $C$ = number of collectors; $h$ = number of hours of collection; and $S$ = number of collection sites.

**Positivity Index of immature forms (PI)** estimated by the formula: $PI = \frac{PBs}{TB}$ where: $PBs$ = number of positive breeding sites for a given species; $TB$ = total number of breeding sites surveyed.

**General Breeding Index (GBI)**, which is a measure of the ratio of water collections that mosquitoes were found developing in a given locality, determined by the equation: $GBI = \frac{PB}{TW}$ where: $PB$ = number of breeding sites with anopheline immatures (positive breeding sites); $TW$ = total number of water collections obtained.

**Absolute Breeding Index (ABI)**, which is the relative ratio of breeding sites occupied by a vector species in a given locality, expressed as: $ABI = \frac{PBs}{NW}$ where: $PBs$ = number of positive breeding sites for a given species; $NW$ = total number of water collections obtained.

**Relative Breeding Index (RBI)**, which reflects the abundance of breeding sites of a given species compared to the number of water collections where mosquitoes were found in a locality, estimated by the formula: $RBI = \frac{PBs}{TPB}$ where: $PBs$ = number of breeding sites positive for a given species; $TPB$ = total number of positive breeding sites.

## Limnological parameters

Of 52 larval habitats, 20 were selected for the limnological analyses. Samples of water were collected in four periods, as follows: two in the dry season (November and December 2019) and two in the rainy season (February and June 2020). Water samples from the larval habitats were collected with the aid of a long dipper (described above) and transferred into sterile 500-mL autoclavable polypropylene bottles. Samples were stored in a thermal box refrigerated at 4°C to 8°C, and transported to the Laboratory of Water Quality Control and Sewage Company of Amapá, where the analyses were conducted on the same day that the samples were collected.

The physicochemical parameters analyzed were: Hydrogen potential (pH), water temperature (°C), dissolved oxygen (DO—mg/L), color, electrical conductivity, total dissolved solids (TDS), nitrogen forms (nitrate, nitrite and ammonia), total phosphate, and turbidity (UNT). Water temperature (°C) and dissolved oxygen—OD (mg/L) of water samples were measured in the field using an *oximeter* (Digimed, model DM-4P). The remaining analyses were carried out in the laboratory. The hydrogen potential was measured using the electrometric method with a benchtop pH meter (Hach—Hexis científica S/A, model sensION^tm PH31). The turbidity analysis was performed using the nephorometric method via a turbidimeter (PoliControl, model AP2000). The water color was assessed according to the APHA Platinum—Cobalt Standard Method (method 8025). The analysis of total phosphate (method 8048), including nitrate (method 8039), nitrite (method 8507), and ammonia (method 8038) was performed using a

spectrophotometer (Hach, model DR 3900). Finally, analyses of electrical conductivity and content of total dissolved solids (STD) were carried out using an electrical conductivity meter (Hanna, model HI 8730). All analyses of physicochemical parameters were performed according to the Standard Methods for the Examination of Water and Wastewater [27] and the CETESB Collection and Sampling Guide [28].

To quantify the environmental carrying capacity of a water body, the main parameters of water quality needed to be evaluated with physicochemical measurements [29]. Water quality parameters were established following the resolutions of the National Council for the Environment-CONAMA (Resolution #357/2005 of March 17th and Resolution #430/2011 of May 13th), which classify water bodies and provide environmental guidelines for their classification, in addition to establishing the conditions and patterns of effluent discharge. The analysis of relevant parameters is described in Art. 42 for the classification of class 2 fresh waters [30].

## Statistical analyses

Generalized Linear Models (GLMs) were applied according to the characteristics and distribution of the dependent variable. The most appropriate GLM for the analysis of environmental parameters was the multinomial logistic regression which evaluates the relationship of categorical structural variables (classification of larval habitat, shading, type of water, presence of debris, current and vegetation) to the abundance of species. GLM-normal distribution was used to examine species abundance and richness relative to the size of the breeding site and the distance to the nearest human settlement. GLM-negative binomial was the most appropriate model to analyze which of the limnological parameters most influenced species occurrence.

A Kruskal-Wallis's test was applied to the non-parametric dataset to evaluate whether differences in abundance were significant, based on structural characteristics and comparisons between breeding sites and water physicochemical factors.

Multivariate analyses (Canonical Correspondence Analysis—CCA) were used to evaluate changes in species composition and response to variations in water physicochemical factors (Fig 5). Tables provide values obtained from GLM models for all variables tested for the collected species, with physicochemical parameter values obtained from all four collections, and with species abundance per breeding site along with entomological parameters.

The data were organized using Excel-Office 2019, with support for statistical analysis. Tests and graphs were generated in the Palaeontological Statistics–Past, version 1.34 [31] and BioEstat, version 5.0 [32]. The significance level was of 95% ($\alpha = 0.05$).

## Ethics statement

The breeding sites were located on private properties and the collection of samples for this study was conducted upon the authorization of the property owners. To maintain the confidentiality of owners, the breeding sites were identified using numbers, as shown in Tables 1 and 3 and Fig 1 of this study. This study obtained a permit from the ethics committee of Federal University of Amapá, under license number: 78912617.9.0000.0003. The collection and transport of anophelines were conceded by the Information and Authorization System on Biodiversity, under license number: 52442–1.

## Results

Fifty-two breeding sites were sampled in Ilha de Santana: 46 fishponds, three ponds, and three streams (Fig 1). Of these, 50 (96.15%) were positive for *Anopheles* larvae and a total of 1,833 specimens were collected. Two breeding sites were negative: a fish farming tank (BS9) and a

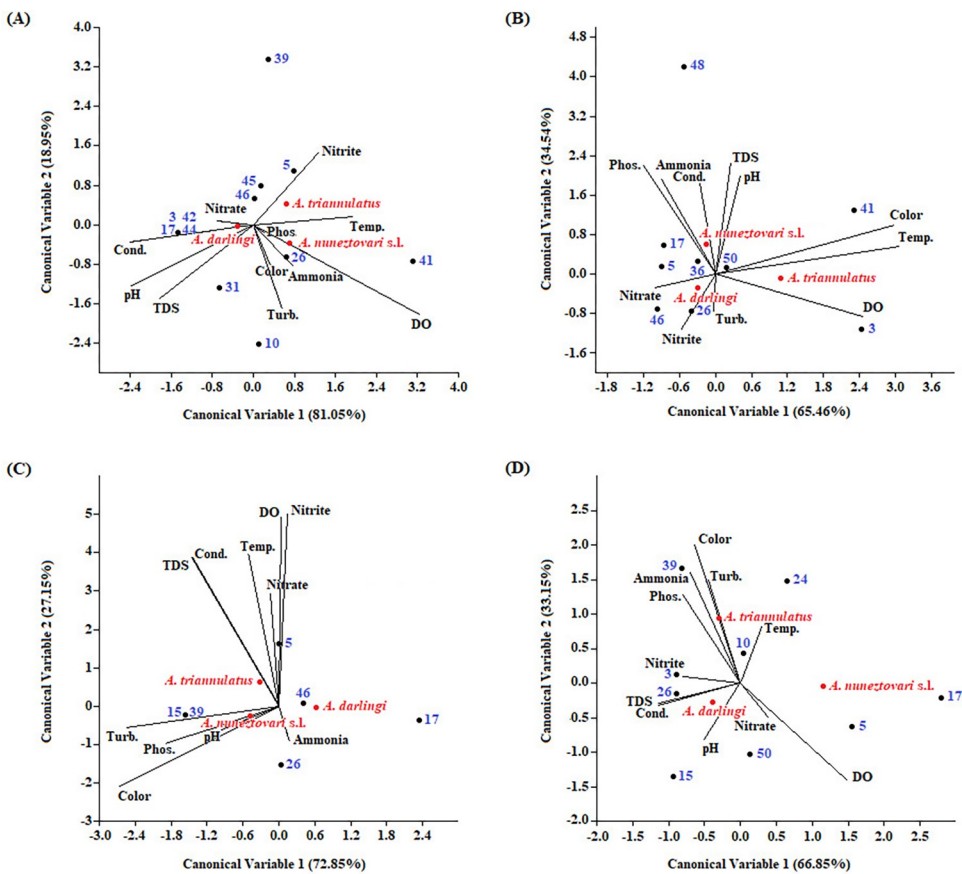

**Fig 5. Canonical Correspondence Analysis (CCA), biplot of ordering diagram showing the dispersion of the three species of *Anopheles* larvae collected in greater abundance and number of breeding sites, and the eleven variables of the limnological parameters contributing to the first two canonical axes. A.** Sample collected in November/2019; **B.** Sample collected in December/2019; **C.** Sample collected in February/2020; **D.** Sample collected in June/2020. **Black spots:** Medium centroid for each breeding site (collection sites); **Red dots:** Medium centroid for *Anopheles* species. In brackets on the axes: Contribution of each canonical variable to the total variation.

stream (BS35**) (Table 1). Nine species were identified, six belong to the *Nyssorhynchus* subgenus (*Anopheles darlingi*, *Anopheles nuneztovari* s.l., *Anopheles triannulatus*, *Anopheles albitarsis* s.l., *Anopheles braziliensis*, and *Anopheles konderi*) and three to the subgenus *Anopheles* (*Anopheles medialis*, *Anopheles mattogrossensis*, and *Anopheles peryassui*). At least two species were collected at each breeding site. *Anopheles darlingi*, *A. nuneztovari* s.l. and *A. triannulatus* were collected at 47 (94%) of the breeding sites, whereas *A. albitarsis* s.l. was sampled at 28 (56%) breeding sites.

## Structural characterization of larval habitats

Of 50 positive larval habitats, 45 (90%) were classified as artificial (fish farming tanks) and five (10%) as natural sites, being three ponds (6%) and two streams (4%). All larval habitats were classified as permanent (Figs 2–4). The fish farming tanks and streams had the highest species richness, both totaling nine species and an abundance of 1,632 (tanks) and 44 (streams) specimens. In ponds, six species were sampled and an abundance of 157 specimens. When analyzing each breeding site separately, species richness was the highest in one stream, with eight species, whereas the highest abundance was found in one pond, with 121 specimens. High

**Table 1. Number of individuals collected per species and entomological parameters (LIMH, PI, GBI, ABI and RBI) estimated from larval habitats in the District of Ilha de Santana, in Brazilian state of Amapá.**

| Breeding | A. albitarsis s.l. | A. braziliensis | A. darlingi | A. medialis | A. konderi | A. mattogrossensis | A. nuneztovari s.l. | A. peryassui | A. triannulatus | Total | LIMH |
|---|---|---|---|---|---|---|---|---|---|---|---|
| BS1 | 7 | 1 | 37 | - | - | - | 28 | - | 15 | 88 | 0.564 |
| BS2 | 10 | 3 | 40 | 2 | 1 | - | 16 | - | 21 | 93 | 0.596 |
| BS3 | 8 | 7 | 27 | - | - | - | 12 | - | 12 | 66 | 0.423 |
| BS4* | 2 | - | 5 | - | - | - | 4 | 2 | 3 | 16 | 0.103 |
| BS5 | 9 | 2 | 25 | - | - | - | 17 | - | 13 | 66 | 0.423 |
| BS6* | 1 | - | 9 | 2 | - | - | 4 | - | 4 | 20 | 0.128 |
| BS7 | 3 | 3 | 14 | - | - | - | 2 | - | 4 | 26 | 0.167 |
| BS8 | - | - | 7 | - | - | - | 5 | - | 6 | 18 | 0.115 |
| BS9 | - | - | - | - | - | - | - | - | - | - | 0.000 |
| BS10 | 2 | - | 12 | - | 2 | - | 3 | - | 5 | 24 | 0.154 |
| BS11 | 2 | - | 15 | - | - | - | 2 | - | 7 | 26 | 0.167 |
| BS12 | 3 | 1 | 5 | - | 5 | - | 7 | 2 | 3 | 26 | 0.167 |
| BS13 | - | 1 | 7 | - | - | - | 7 | - | 7 | 22 | 0.141 |
| BS14 | 2 | - | 2 | 1 | - | - | 2 | - | 3 | 10 | 0.064 |
| BS15 | 2 | 1 | 12 | - | - | - | 7 | - | 6 | 28 | 0.179 |
| BS16 | 2 | - | 4 | - | - | - | 3 | - | 4 | 13 | 0.083 |
| BS17 | - | - | 17 | - | 12 | 1 | 14 | 1 | 6 | 51 | 0.327 |
| BS18 | - | - | - | - | 2 | - | - | 1 | 2 | 5 | 0.032 |
| BS19 | 1 | - | 12 | - | - | - | 6 | - | 3 | 22 | 0.141 |
| BS20 | - | 2 | 6 | - | - | - | 8 | - | 3 | 19 | 0.122 |
| BS21 | - | - | 9 | - | - | - | 8 | - | 6 | 23 | 0.147 |
| BS22 | 1 | - | 6 | 1 | - | 1 | 2 | - | 5 | 16 | 0.103 |
| BS23 | - | - | 1 | - | - | - | 6 | 2 | 3 | 12 | 0.077 |
| BS24 | 1 | 2 | 11 | - | 2 | - | 14 | - | 11 | 41 | 0.263 |
| BS25 | 9 | 5 | 32 | - | 1 | - | 16 | - | 13 | 76 | 0.487 |
| BS26* | 3 | 12 | 56 | 1 | - | - | 30 | - | 19 | 121 | 0.776 |
| BS27 | 2 | 3 | 16 | - | 4 | - | 8 | - | 2 | 35 | 0.224 |
| BS28 | 5 | 2 | 20 | 2 | 5 | - | 2 | - | 4 | 40 | 0.256 |
| BS29 | 2 | - | 22 | - | - | - | 11 | - | 4 | 39 | 0.250 |
| BS30 | - | 4 | 19 | - | 8 | - | 3 | 2 | 6 | 42 | 0.269 |
| BS31 | - | 3 | 10 | - | - | - | 2 | - | - | 15 | 0.096 |
| BS32 | - | - | 12 | - | 6 | - | 2 | 2 | 4 | 26 | 0.167 |
| BS33 | - | 1 | 12 | - | - | - | 3 | - | 2 | 18 | 0.115 |
| BS34 | 3 | - | 10 | - | - | - | - | - | 4 | 17 | 0.109 |
| BS35** | - | - | - | - | - | - | - | - | - | - | 0.000 |
| BS36 | - | 2 | 7 | - | 1 | - | 6 | - | 2 | 18 | 0.115 |
| BS37 | - | - | 8 | - | 3 | - | 3 | - | - | 14 | 0.090 |
| BS38 | 1 | - | 17 | 4 | 7 | - | 18 | - | 19 | 66 | 0.423 |
| BS39 | - | 2 | 13 | - | 4 | - | 21 | 2 | 19 | 61 | 0.391 |
| BS40 | - | 1 | 8 | - | - | - | 14 | - | 14 | 37 | 0.237 |
| BS41 | - | - | 2 | 1 | - | - | 12 | 2 | 9 | 26 | 0.167 |
| BS42 | 4 | - | 11 | - | 3 | - | 2 | 2 | 15 | 37 | 0.237 |
| BS43** | 4 | 2 | 12 | 2 | 1 | 5 | - | 2 | 7 | 35 | 0.224 |
| BS44 | - | 3 | 14 | - | - | - | 3 | 2 | 9 | 31 | 0.199 |
| BS45 | 6 | - | 20 | - | 4 | - | 21 | 1 | 8 | 60 | 0.385 |
| BS46 | - | 4 | 48 | 2 | 3 | - | 25 | - | 19 | 101 | 0.647 |

(*Continued*)

**Table 1.** (Continued)

| Breeding | A. albitarsis s.l. | A. braziliensis | A. darlingi | A. medialis | A. konderi | A. mattogrossensis | A. nuneztovari s.l. | A. peryassui | A. triannulatus | Total | LIMH |
|---|---|---|---|---|---|---|---|---|---|---|---|
| BS47 | - | 2 | 14 | - | - | - | 4 | - | 5 | 25 | 0.160 |
| BS48 | - | - | - | - | 3 | - | 3 | - | - | 6 | 0.038 |
| BS49** | - | - | - | 1 | - | - | 4 | 2 | 2 | 9 | 0.058 |
| BS50 | - | 2 | 32 | - | 2 | - | 18 | - | 10 | 64 | 0.410 |
| BS51 | 3 | 2 | 15 | 2 | 1 | - | 16 | - | 3 | 42 | 0.269 |
| BS52 | 3 | - | 19 | - | 5 | - | 10 | 2 | 2 | 41 | 0.263 |
| Total | 101 | 73 | 732 | 21 | 85 | 7 | 434 | 27 | 353 | 1.833 | |
| LIMH | 0.027 | 0.019 | 0.196 | 0.006 | 0.023 | 0.002 | 0.116 | 0.007 | 0.094 | 0.490 | |
| PI | 0.538 (53.80%) | 0.500 (50.00%) | 0.904 (90.40%) | 0.231 (23.10%) | 0.442 (44.20%) | 0.058 (5.80%) | 0.904 (90.40%) | 0.289 (28.90%) | 0.904 (90.40%) | | |
| GBI | | | | | | | | | | 0.048 | |
| ABI | 0.019 | 0.017 | 0.032 | 0.008 | 0.015 | 0.002 | 0.032 | 0.010 | 0.032 | | |
| RBI | 0.560 | 0.520 | 0.940 | 0.240 | 0.460 | 0.060 | 0.940 | 0.300 | 0.940 | | |

*: Pond

**: Stream; Fish farming tanks are indicated without an asterisk. **LIMH**: Larvae Index per Man/Hour; **PI**: Positivity Index; **GBI**: General Breeding Index; **ABI**: Absolute Breeding Index; **RBI**: Relative Breeding Index. -: Negative breeding sites for Anophelines.

species richness and abundance were also observed in fish farming tanks, with up to seven species and 101 individuals. The ponds exhibited structural characteristics similar to those of fish farming tanks, whereas streams showed differences regarding the water type (clear) and the current (moderate).

**Table 2. Generalized Linear Models of the environmental and limnological parameters of the larval habitats sampled in the District of Ilha de Santana, Brazilian state of Amapá.**

| Parameters | Species | GLM multinomial logistics | | | |
|---|---|---|---|---|---|
| | | β | SE | G value | p value |
| Classification (fish farming tanks, lagoons and streams) | A. albitarsis s.l. | -0.058 | 0.549 | 0.011 | 0.915 |
| | A. braziliensis | -0.467 | 0.584 | 0.684 | 0.408 |
| | A. darlingi | -1.508 | 0.642 | 5.256 | 0.021* |
| | A. medialis | 1.358 | 0.636 | 5.346 | 0.020* |
| | A. konderi | -0.693 | 0.670 | 1.282 | 0.257 |
| | A. mattogrossensis | 1.112 | 0.733 | 1.897 | 0.168 |
| | A. nuneztovari s.l. | -1.508 | 0.642 | 5.256 | 0.021* |
| | A. peryassui | 0.771 | 0.566 | 1.89 | 0.169 |
| | A. triannulatus | -0.679 | 0.681 | 0.848 | 0.356 |
| Shading | A. albitarsis s.l. | 0.030 | 0.632 | 0.002 | 0.961 |
| | A. braziliensis | -0.393 | 0.630 | 0.392 | 0.531 |
| | A. darlingi | -16.679 | 1,519.700 | 4.379 | 0.036* |
| | A. medialis | -0.405 | 0.712 | 0.316 | 0.574 |
| | A. konderi | 0.076 | 0.630 | 0.014 | 0.903 |
| | A. mattogrossensis | -0.325 | 1.266 | 0.063 | 0.801 |
| | A. nuneztovari s.l. | -16.679 | 1,519.700 | 4.379 | 0.036* |
| | A. peryassui | -0.441 | 0.668 | 0.429 | 0.512 |
| | A. triannulatus | -16.679 | 1,519.70 | 4.379 | 0.036* |

(Continued)

**Table 2.** (Continued)

| Water nature | *A. albitarsis* s.l. | 0.818 | 0.358 | 6.066 | 0.013* |
|---|---|---|---|---|---|
| | *A. braziliensis* | 0.289 | 0.315 | 0.862 | 0.353 |
| | *A. darlingi* | 1.571 | 0.595 | 9.075 | 0.002* |
| | *A. medialis* | 0.332 | 0.397 | 0.746 | 0.387 |
| | *A. konderi* | -0.348 | 0.317 | 1.236 | 0.266 |
| | *A. mattogrossensis* | 0.929 | 0.909 | 1.306 | 0.253 |
| | *A. nuneztovari* s.l. | 0.267 | 0.494 | 0.282 | 0.595 |
| | *A. peryassui* | -0.351 | 0.334 | 1.100 | 0.294 |
| | *A. triannulatus* | 0.985 | 0.503 | 4.026 | 0.044* |
| Debris | *A. albitarsis* s.l. | -0.172 | 0.219 | 0.627 | 0.428 |
| | *A. braziliensis* | -0.268 | 0.233 | 1.435 | 0.230 |
| | *A. darlingi* | -0.277 | 0.304 | 0.751 | 0.386 |
| | *A. medialis* | -0.482 | 0.356 | 2.336 | 0.126 |
| | *A. konderi* | -0.177 | 0.228 | 0.633 | 0.426 |
| | *A. mattogrossensis* | -0.221 | 0.556 | 0.179 | 0.671 |
| | *A. nuneztovari* s.l. | -0.453 | 0.292 | 2.216 | 0.136 |
| | *A. peryassui* | -0.140 | 0.255 | 0.318 | 0.572 |
| | *A. triannulatus* | -0.721 | 0.312 | 5.723 | 0.016* |
| Flow | *A. albitarsis* s.l. | 0.215 | 0.457 | 0.228 | 632.000 |
| | *A. braziliensis* | -0.390 | 0.457 | 0.766 | 0.381 |
| | *A. darlingi* | -0.984 | 0.525 | 3.102 | 0.078 |
| | *A. medialis* | 0.293 | 0.468 | 0.373 | 0.541 |
| | *A. konderi* | -0.259 | 0.457 | 0.334 | 0.563 |
| | *A. mattogrossensis* | 0.740 | 0.656 | 1.078 | 0.299 |
| | *A. nuneztovari* s.l. | -1.545 | 0.545 | 8.149 | 0.004* |
| | *A. peryassui* | 0.520 | 0.441 | 1.357 | 0.244 |
| | *A. triannulatus* | -0.370 | 0.607 | 0.334 | 0.563 |
| Vegetation | *A. albitarsis* s.l. | -0.080 | 0.366 | 0.049 | 0.824 |
| | *A. braziliensis* | -0.127 | 0.360 | 0.126 | 0.722 |
| | *A. darlingi* | -15.528 | 1,125.90 | 3.474 | 0.062 |
| | *A. medialis* | 0.241 | 0.502 | 0.261 | 0.609 |
| | *A. konderi* | 0.270 | 0.369 | 0.512 | 0.474 |
| | *A. mattogrossensis* | 0.022 | 0.778 | 0.001 | 0.976 |
| | *A. nuneztovari* s.l. | 0.341 | 0.462 | 0.471 | 0.492 |
| | *A. peryassui* | -0.056 | 0.177 | 0.102 | 0.784 |
| | *A. triannulatus* | -15.528 | 1,258.80 | 2.779 | 0.095 |
| **Parameters** | **Species** | **GLM negative binomial model** | | | |
| | | *β* | *SE* | *G value* | *p value* |
| pH | *A. braziliensis* | -1.302 | 1.268 | 0.581 | 0.445 |
| | *A. darlingi* | 3.733 | 1.234 | 5.685 | 0.017* |
| | *A. konderi* | -0.532 | 1.770 | 0.071 | 0.790 |
| | *A. nuneztovari* s.l. | 0.296 | 0.556 | 0.320 | 0.571 |
| | *A. triannulatus* | 1.093 | 0.513 | 5.453 | 0.019* |
| Temperature | *A. braziliensis* | 0.573 | 0.543 | 0.774 | 0.378 |
| | *A. darlingi* | 0.631 | 0.703 | 0.534 | 0.464 |
| | *A. konderi* | -0.472 | 0.474 | 1.551 | 0.213 |
| | *A. nuneztovari* s.l. | -0.099 | 0.258 | 0.149 | 0.699 |
| | *A. triannulatus* | 0.518 | 0.203 | 5.120 | 0.023* |

(*Continued*)

**Table 2.** (Continued)

| | | β | SE | | |
|---|---|---|---|---|---|
| Dissolved oxygen | *A. braziliensis* | -0.683 | 1.046 | 0.527 | 0.467 |
| | *A. darlingi* | 0.004 | 1.226 | 0.009 | 0.997 |
| | *A. konderi* | -0.403 | 1.721 | 0.028 | 0.865 |
| | *A. nuneztovari* s.l. | -0.020 | 0.334 | 0.005 | 0.944 |
| | *A. triannulatus* | 0.493 | 0.349 | 2.559 | 0.109 |
| Nitrate | *A. braziliensis* | -2.910 | 1.426 | 3.972 | 0.046* |
| | *A. darlingi* | -0.893 | 1.651 | 0.230 | 0.630 |
| | *A. konderi* | -1.562 | 1.119 | 1.221 | 0.269 |
| | *A. nuneztovari* s.l. | 0.102 | 0.775 | 0.017 | 0.894 |
| | *A. triannulatus* | -0.615 | 0.520 | 0.573 | 0.449 |
| Ammonia | *A. braziliensis* | 4.497 | 3.545 | 1.285 | 0.257 |
| | *A. darlingi* | 6.742 | 2.535 | 5.921 | 0.014* |
| | *A. konderi* | 2.346 | 4.141 | 0.045 | 0.831 |
| | *A. nuneztovari* s.l. | 1.601 | 1.005 | 2.894 | 0.088 |
| | *A. triannulatus* | 2.491 | 0.997 | 4.957 | 0.025* |
| Turbidity | *A. braziliensis* | 0.031 | 0.021 | 4.052 | 0.044* |
| | *A. darlingi* | 0.009 | 0.026 | 0.236 | 0.627 |
| | *A. konderi* | -0.025 | 0.017 | 1.409 | 0.235 |
| | *A. nuneztovari* s.l. | -0.001 | 0.005 | 1.254 | 0.965 |
| | *A. triannulatus* | -0.007 | 0.002 | 0.195 | 0.262 |
| Color | *A. braziliensis* | 0.005 | 0.006 | 0.437 | 0.508 |
| | *A. darlingi* | -0.006 | 0.004 | 0.838 | 0.359 |
| | *A. konderi* | -1.096 | 0.016 | 0.251 | 0.616 |
| | *A. nuneztovari* s.l. | 0.001 | 0.002 | 0.011 | 0.917 |
| | *A. triannulatus* | 0.001 | 0.002 | 0.195 | 0.658 |
| Phosphate | *A. braziliensis* | 5.906 | 4.702 | 1.022 | 0.312 |
| | *A. darlingi* | 7.851 | 5.453 | 2.940 | 0.086 |
| | *A. konderi* | -1.456 | 2.429 | 0.448 | 0.503 |
| | *A. nuneztovari* s.l. | 2.441 | 1.812 | 2.426 | 0.119 |
| | *A. triannulatus* | 2.324 | 2.292 | 1.170 | 0.191 |
| Total dissolved solids | *A. braziliensis* | 0.026 | 0.218 | 0.022 | 0.879 |
| | *A. darlingi* | 0.078 | 0.112 | 0.451 | 0.501 |
| | *A. konderi* | -0.102 | 0.126 | 0.756 | 0.384 |
| | *A. nuneztovari* s.l. | 0.002 | 0.043 | 0.002 | 0.963 |
| | *A. triannulatus* | 0.006 | 0.041 | 0.017 | 0.895 |
| Eletrical conductivity | *A. braziliensis* | -0.013 | 0.114 | 0.022 | 0.880 |
| | *A. darlingi* | 0.039 | 0.059 | 0.488 | 0.484 |
| | *A. konderi* | -0.057 | 0.064 | 1.016 | 0.313 |
| | *A. nuneztovari* s.l. | 0.003 | 0.022 | 0.024 | 0.844 |
| | *A. triannulatus* | 0.019 | 0.023 | 0.560 | 0.454 |
| Nitrite | *A. braziliensis* | 500.000 | 864.520 | 0.389 | 0.532 |
| | *A. darlingi* | 777.780 | 1.420.900 | 393.000 | 0.530 |
| | *A. konderi* | - | - | - | - |
| | *A. nuneztovari* s.l. | -265.150 | 379.220 | 0.399 | 0.527 |
| | *A. triannulatus* | 186.270 | 445.180 | 0.202 | 0.653 |

β: estimation; SE: standard error

* significant at $p < 0.05$; (-) Nitrite values equal to 0.000 for positive breeding sites of *A. konderi*.

**Table 3. Limnological parameters analyzed in the four water samples collected from 20 breeding sites in the District of Ilha de Santana, Brazilian state of Amapá.**

| | Breeding sites analyzed | pH | Temperature (°C) | Nitrate (NO$_3^-$-N) | Ammonia (NH$_3$-N) | DO (mg/L) | Turbidity (NTU) | Color | Phosphate (PO$_4^{-3}$) | TDS (ppm) | Eletrical conductivity (μS/cm) | Nitrite (NO$_2^-$-N) |
|---|---|---|---|---|---|---|---|---|---|---|---|---|
| 1ª. Sample | 3 | 6.90 | 29.7 | 0.5 | 0.39 | 1.20 | 14.50 | 217 | 0.16 | 10.00 | 30.0 | 0.000 |
| | 5 | 6.21 | 28.9 | 0.5 | 0.32 | 1.46 | 8.21 | 131 | 0.15 | 0.00 | 20.0 | 0.002 |
| | 6 | 6.16 | 27.8 | 0.0 | 0.24 | 1.05 | 11.00 | 159 | 0.43 | 10.00 | 30.0 | 0.000 |
| | 10 | 6.28 | 28.9 | 0.0 | 0.46 | 2.11 | 23.60 | 287 | 0.15 | 20.00 | 40.0 | 0.000 |
| | 14 | 6.21 | 29.5 | 0.0 | 0.30 | 1.09 | 8.26 | 135 | 0.29 | 10.00 | 30.0 | 0.000 |
| | 15 | 6.22 | 27.1 | 0.0 | 0.48 | 1.26 | 28.80 | 337 | 0.51 | 20.00 | 40.0 | 0.000 |
| | 17 | 6.28 | 28.3 | 0.3 | 0.25 | 1.11 | 7.49 | 127 | 0.28 | 20.00 | 40.0 | 0.000 |
| | 18 | 6.25 | 28.7 | 2.3 | 1.70 | 2.74 | 133.00 | 1305 | 0.66 | 10.00 | 30.0 | 0.000 |
| | 24 | 6.25 | 28.8 | 0.0 | 2.20 | 0.45 | 152.00 | 1640 | 0.70 | 20.00 | 40.0 | 0.000 |
| | 26 | 5.86 | 28.4 | 0.6 | 0.48 | 0.35 | 8.32 | 234 | 0.47 | 10.00 | 20.0 | 0.000 |
| | 31 | 5.72 | 27.1 | 0.4 | 0.44 | 0.81 | 7.57 | 162 | 0.53 | 10.00 | 20.0 | 0.000 |
| | 36 | 5.61 | 28.4 | 4.1 | 0.15 | 1.46 | 2.00 | 89 | 0.35 | 40.00 | 80.0 | 0.003 |
| | 39 | 5.79 | 28.8 | 0.1 | 0.45 | 0.84 | 8.69 | 262 | 0.38 | 10.00 | 30.0 | 0.000 |
| | 41 | 6.00 | 31.1 | 0.0 | 0.62 | 3.89 | 40.70 | 415 | 0.16 | 10.00 | 30.0 | 0.000 |
| | 42 | 6.14 | 30.6 | 0.0 | 0.86 | 1.72 | 57.40 | 661 | 0.05 | 10.00 | 40.0 | 0.000 |
| | 44 | 6.20 | 29.0 | 0.1 | 0.34 | 1.80 | 17.30 | 220 | 0.13 | 20.00 | 50.0 | 0.000 |
| | 45 | 5.92 | 29.2 | 0.3 | 0.24 | 1.21 | 6.80 | 91 | 0.06 | 20.00 | 40.0 | 0.000 |
| | 46 | 5.95 | 28.6 | 0.0 | 0.55 | 1.10 | 5.92 | 252 | 0.05 | 20.00 | 40.0 | 0.000 |
| | 48 | 6.12 | 28.5 | 0.0 | 0.46 | 0.67 | 14.00 | 243 | 0.15 | 20.00 | 50.0 | 0.000 |
| | 50 | 6.24 | 28.1 | 0.0 | 0.33 | 2.16 | 18.90 | 210 | 0.09 | 20.00 | 40.0 | 0.000 |
| 2ª. Sample | 3 | 5.84 | 29.4 | 0.0 | 0.44 | 1.88 | 22.40 | 328 | 0.20 | 20.00 | 40.0 | 0.000 |
| | 5 | 6.17 | 28.3 | 0.6 | 0.56 | 1.36 | 13.30 | 160 | 0.20 | 10.00 | 20.0 | 0.000 |
| | 6 | 6.36 | 27.6 | 0.2 | 0.50 | 4.36 | 13.90 | 179 | 0.21 | 20.00 | 50.0 | 0.001 |
| | 10 | 6.31 | 28.0 | 0.0 | 0.47 | 2.01 | 10.30 | 129 | 0.21 | 20.00 | 50.0 | 0.000 |
| | 14 | 6.46 | 28.4 | 0.1 | 0.51 | 1.86 | 30.50 | 355 | 0.47 | 20.00 | 50.0 | 0.000 |
| | 15 | 6.49 | 26.7 | 0.0 | 0.48 | 1.49 | 11.60 | 152 | 0.28 | 20.00 | 50.0 | 0.000 |
| | 17 | 6.34 | 27.2 | 0.2 | 0.50 | 1.01 | 12.10 | 155 | 0.49 | 20.00 | 50.0 | 0.000 |
| | 18 | 6.47 | 28.3 | 0.0 | 0.44 | 3.75 | 78.20 | 808 | 0.82 | 20.00 | 50.0 | 0.000 |
| | 24 | 6.36 | 28.1 | 0.0 | 0.49 | 3.08 | 72.30 | 734 | 0.49 | 20.00 | 50.0 | 0.000 |
| | 26 | 5.86 | 28.4 | 0.0 | 0.71 | 0.19 | 136.00 | 209 | 0.28 | 10.00 | 20.0 | 0.002 |
| | 31 | 6.17 | 27.6 | 0.2 | 0.54 | 0.27 | 3.79 | 145 | 0.29 | 10.00 | 20.0 | 0.000 |
| | 36 | 5.83 | 27.6 | 1.6 | 0.56 | 1.00 | 113.00 | 34 | 0.20 | 30.00 | 60.0 | 0.000 |
| | 39 | 7.02 | 30.7 | 0.0 | 0.68 | 4.49 | 35.60 | 392 | 0.18 | 20.00 | 40.0 | 0.000 |
| | 41 | 7.30 | 30.1 | 0.0 | 0.74 | 2.97 | 49.60 | 509 | 0.30 | 20.00 | 40.0 | 0.000 |
| | 42 | 7.09 | 29.8 | 0.1 | 0.76 | 1.59 | 6.85 | 88 | 0.43 | 10.00 | 30.0 | 0.000 |
| | 44 | 6.50 | 28.8 | 0.4 | 0.75 | 1.14 | 7.06 | 98 | 0.29 | 20.00 | 50.0 | 0.000 |
| | 45 | 6.86 | 28.4 | 0.0 | 0.90 | 2.41 | 13.08 | 161 | 0.34 | 30.00 | 60.0 | 0.000 |
| | 46 | 6.88 | 28.6 | 0.0 | 0.92 | 1.74 | 9.86 | 208 | 0.31 | 20.00 | 50.0 | 0.000 |
| | 48 | 6.96 | 28.9 | 0.0 | 1.05 | 0.39 | 21.50 | 293 | 0.46 | 30.00 | 60.0 | 0.000 |
| | 50 | 7.09 | 28.5 | 0.0 | 1.06 | 2.62 | 17.10 | 214 | 0.42 | 20.00 | 50.0 | 0.000 |

(*Continued*)

**Table 3.** (Continued)

| | Breeding sites analyzed | pH | Temperature (°C) | Nitrate (NO₃-N) | Ammonia (NH₃-N) | DO (mg/L) | Turbidity (NTU) | Color | Phosphate (PO₄⁻³) | TDS (ppm) | Eletrical conductivity (µS/cm) | Nitrite (NO₂-N) |
|---|---|---|---|---|---|---|---|---|---|---|---|---|
| 3ª. Sample | **3** | **6.20** | 27.6 | **0.2** | **0.25** | 1.13 | **6.39** | 145 | **0.01** | **20.00** | 40.0 | **0.000** |
| | **5** | 5.60 | 28.1 | **0.8** | **0.22** | 2.28 | **6.69** | **40** | 0.00 | **15.00** | 30.0 | **0.002** |
| | **6** | 5.48 | 26.4 | **0.5** | **0.54** | 0.99 | **5.97** | 138 | 0.07 | **5.00** | 10.0 | **0.000** |
| | **10** | **6.05** | 26.8 | **0.0** | **0.66** | 0.96 | **28.00** | 322 | 0.13 | **20.00** | 40.0 | **0.000** |
| | **14** | 5.48 | 27.2 | **0.0** | **0.32** | 1.21 | **57.10** | 500 | 0.29 | **10.00** | 20.0 | **0.000** |
| | **15** | 5.42 | 27.5 | **0.1** | **0.20** | 1.03 | **13.80** | 168 | 0.07 | **10.00** | 20.0 | **0.000** |
| | **17** | 5.60 | 26.8 | **0.0** | **0.53** | 1.10 | **11.10** | 142 | 0.06 | **10.00** | 20.0 | **0.000** |
| | **18** | 5.58 | 27.8 | **0.0** | **0.60** | 1.56 | **31.40** | 345 | 0.16 | **15.00** | 30.0 | **0.000** |
| | **24** | **6.11** | 27.2 | **0.0** | **0.30** | 1.96 | **30.40** | 366 | 0.15 | **20.00** | 40.0 | **0.000** |
| | **26** | 5.65 | 26.6 | **0.4** | **0.25** | 0.58 | **6.65** | 138 | **0.02** | **10.00** | 20.0 | **0.000** |
| | **31** | 5.57 | 26.0 | **0.1** | **0.26** | 0.52 | **7.56** | 159 | 0.43 | **10.00** | 20.0 | **0.000** |
| | **36** | 5.08 | 27.9 | **2.8** | **0.03** | 1.71 | **0.10** | **2** | **0.02** | **45.00** | 90.0 | **0.003** |
| | **39** | 5.91 | 26.3 | **0.0** | **0.73** | 0.99 | **16.60** | 352 | 0.15 | **15.00** | 30.0 | **0.000** |
| | **41** | 5.52 | 27.8 | **0.0** | **0.42** | 1.04 | **23.00** | 238 | 0.04 | **15.00** | 30.0 | **0.000** |
| | **42** | 5.80 | 26.8 | **0.0** | **0.68** | 0.46 | **15.00** | 382 | 0.14 | **15.00** | 30.0 | **0.000** |
| | **44** | **6.13** | 26.3 | **0.0** | **0.48** | 1.10 | **19.70** | 253 | 0.13 | **25.00** | 50.0 | **0.000** |
| | **45** | 5.74 | 27.1 | **0.1** | **0.24** | 1.20 | **11.80** | 134 | **0.03** | **10.00** | 20.0 | **0.000** |
| | **46** | 5.61 | 26.3 | **0.0** | **0.31** | 0.47 | **12.70** | 206 | 0.12 | **15.00** | 30.0 | **0.000** |
| | **48** | **6.17** | 26.6 | **0.0** | **0.60** | 0.48 | **31.60** | 351 | 0.25 | **25.00** | 50.0 | **0.000** |
| | **50** | **6.12** | 27.0 | **0.0** | **0.44** | 2.05 | **23.80** | 273 | 0.93 | **20.00** | 40.0 | **0.000** |
| 4ª. Sample | **3** | 5.57 | 28.4 | **0.7** | **0.08** | 0.77 | **8.76** | 102 | 0.10 | **24.80** | 49.6 | **0.000** |
| | **5** | 5.19 | 27.5 | **0.9** | **0.00** | 0.83 | **5.23** | **53** | 0.10 | **9.30** | 18.6 | **0.000** |
| | **6** | 5.85 | 27.1 | **0.0** | **0.18** | 0.77 | **9.47** | 157 | 0.24 | **15.60** | 31.2 | **0.000** |
| | **10** | **6.09** | 27.3 | **0.0** | **0.30** | 0.80 | **24.90** | 277 | 0.20 | **18.80** | 37.6 | **0.000** |
| | **14** | 5.04 | 28.1 | **0.7** | **0.00** | 0.77 | **6.54** | **62** | 0.15 | **9.85** | 19.7 | **0.001** |
| | **15** | 5.81 | 27.9 | **0.0** | **0.33** | 0.80 | **27.10** | 203 | 0.13 | **12.50** | 25.0 | **0.000** |
| | **17** | 5.21 | 28.1 | **0.0** | **0.10** | 0.79 | **12.60** | 130 | 0.10 | **8.00** | 16.0 | **0.000** |
| | **18** | 5.21 | 29.1 | **0.4** | **0.00** | 0.81 | **1.64** | **40** | 0.08 | **6.30** | 12.6 | **0.000** |
| | **24** | 5.80 | 29.6 | **0.0** | **0.51** | 0.81 | **44.50** | 433 | 0.12 | **10.85** | 21.7 | **0.000** |
| | **26** | 5.20 | 27.6 | **0.0** | **0.02** | 0.77 | **9.31** | 150 | 0.07 | **8.00** | 16.0 | **0.000** |
| | **31** | 5.28 | 26.7 | **0.0** | **0.15** | 0.84 | **21.20** | 192 | 0.15 | **10.00** | 20.0 | **0.000** |
| | **36** | 5.33 | 28.4 | **3.5** | **0.00** | 0.76 | **6.87** | **64** | 0.28 | **45.20** | 90.4 | **0.002** |
| | **39** | 5.00 | 27.8 | **0.0** | **0.40** | 0.71 | **28.80** | 336 | 0.26 | **9.65** | 19.3 | **0.000** |
| | **41** | 5.62 | 27.5 | **0.0** | **0.15** | 0.79 | **16.00** | 188 | 0.20 | **13.80** | 27.6 | **0.000** |
| | **42** | 5.68 | 27.7 | **0.0** | **0.16** | 0.82 | **8.62** | 152 | 0.20 | **13.15** | 26.3 | **0.000** |
| | **44** | 5.79 | 28.2 | **0.0** | **0.30** | 0.71 | **19.00** | 231 | 0.33 | **23.95** | 47.9 | **0.000** |
| | **45** | **6.13** | 27.3 | **0.0** | **0.26** | 0.78 | **19.00** | 223 | 0.63 | **24.85** | 49.7 | **0.000** |
| | **46** | **6.15** | 27.0 | **0.0** | **0.18** | 0.80 | **20.00** | 208 | 0.24 | **8.85** | 17.7 | **0.000** |
| | **48** | 5.38 | 27.6 | **0.0** | **0.30** | 0.73 | **18.00** | 235 | 0.36 | **22.20** | 44.4 | **0.000** |
| | **50** | **6.18** | 28.7 | **0.0** | **0.22** | 0.79 | **18.80** | 226 | 0.17 | **18.00** | 36.0 | **0.000** |

In **bold:** Physical-chemical parameters that are in accordance with the reference values established by CONAMA.

The mean size of fish farming tanks was 44.43 m² (ranging from 25 to 100 m²). The mean size of ponds was 73.33 m² (60 to 80 m²), while streams had a mean size of 30 m² (25 to 35 m²). The mean distance between breeding sites and human settlements was 42.96 m (25 to 75

m). Of these, 20 (40%) breeding sites were located a distance between 25 and 35 m from human settlements, 19 (38%) between 40 and 50 m, and 11 (22%) between 55 and 75 m.

The GLM-normal distribution revealed a significant negative correlation between the abundance of *A. darlingi* (G = 324.870; $\beta$ = -0.017; $p < 0.001$), *A. konderi* (G = 16.679; $\beta$ = -0.030; $p < 0.001$), *A. nuneztovari* s.l. (G = 251.920; $\beta$ = -0.027; $p < 0.001$), and *A. triannulatus* (G = 45.844; $\beta$ = -0.012; $p < 0.001$) and the distance from breeding sites to the nearest human settlement. Similarly, when analyzing species abundance and breeding site size, a negative correlation was observed for: *A. medialis* [= *A. intermedius*] (G = 4.186; $\beta$ = -0.332; $p = 0.003$), *A. konderi* (G = 35.830; $\beta$ = -0.036; $p < 0.001$), and *A. mattogrossensis* (G = 7.391; $\beta$ = -1.846; $p = 0.006$). However, for *A. braziliensis* (G = 4.280; $\beta$ = 0.011; $p < 0.001$), *A. darlingi* (G = 60.815; $\beta$ = 0.004; $p < 0.001$), *A. nuneztovari* s.l. (G = 9.478; $\beta$ = 0.003; $p = 0.002$), and *A. triannulatus* (G = 4.146; $\beta$ = 0.002; $p = 0.041$) a positive correlation was found between abundance and breeding sites size. For species richness, a negative correlation was obtained only for the distance to the nearest settlement (G = 11.994; $\beta$ = -0.008; $p < 0.001$).

The GML-multinomial logistic regression revealed a significant effect of the variables type of breeding site (G = 5.256; $\beta$ = -1.508; $p = 0.021$), shading (G = 4.379; $\beta$ = -16.679; $p = 0.036$), and especially the type of water (G = 9.075; $\beta$ = 1.571; $p = 0.002$) for *A. darlingi* abundance. The abundance of *A. nuneztovari* s.l. was influenced by the following variables: type of breeding site (G = 5.256; $\beta$ = -1.508; $p = 0.021$), shading (G = 4.379; $\beta$ = -16.679; $p = 0.036$), and current (G = 8.149; $\beta$ = -1.545; $p = 0.004$). *Anopheles triannulatus* was affected by the environmental variables: shading (G = 4.379; $\beta$ = -16.679; $p = 0.036$), type of water (G = 4.026; $\beta$ = 0.985; $p = 0.044$), and the presence of debris (G = 5.723; $\beta$ = -0.721; $p = 0.016$). *Anopheles albitarsis* s.l. was affected only by the type of water (G = 6.066; $\beta$ = 0.818; $p = 0.013$) and *A. medialis* by the type of the breeding site (G = 5.256; $\beta$ = -1.508; $p = 0.021$) (Table 2). According to the Kruskal-Wallis's test, the highest abundance of anophelines was observed at breeding sites with shading greater than 50%, polluted or murky waters with debris, no current, and the presence of floating vegetation ($p < 0.001$). The aquatic vegetation found at breeding sites was identified as *Lemma* sp., *Pistia* sp., *Eichornia* sp., *Nymphaea* sp., and *Juncus* sp.

## Entomological parameters

The Larvae Index per Man/Hour (LIMH) was the highest for *A. darlingi* (0.196) and *A. nuneztovari* s.l. (0.116) (Table 1). The highest LIMH was obtained for a fish farming tank (0.776) and the lowest for a pond (0.032). A high positivity index (PI) was found for *A. darlingi*, *A. nuneztovari* s.l., and *A. triannulatus* (0.904). The ratio of water collections used by anopheline species as breeding sites in the study area was 0.048. Despite the low absolute breeding index obtained for *A. darlingi*, *A. nuneztovari* s.l., and *A. triannulatus* (ABI = 0.032), the relative ratio of breeding sites occupied by these species was high, as well as their abundance at breeding sites in relation to the number of water collections (RBI = 0.940) (Table 1).

## Limnological parameters

Four water collections carried out at the 20 breeding sites, totaling 80 samples and these were analyzed for eleven physicochemical parameters. According to the CONAMA Environmental Resolution #357/2005, which establishes standards for the maintenance of aquatic life in natural water bodies, of nine parameters analyzed (reference values for temperature and electrical conductivity are not established by CONAMA), only four (44.45%) were within reference standards (ammonia, nitrate, nitrite and total dissolved solids), three (33.33%) (pH, color, and turbidity) were partially in accordance (only for some breeding sites) and two (22.22%) (dissolved oxygen and phosphate) were out of the range of reference values (Table 3).

Considering all breeding sites together, mean water temperature was 28.03˚C, ranging from 26.00˚C to 31.10˚C, and pH ranging from 5.00 to 7.30. However, no *Anopheles* larvae were collected at breeding sites with pH between 5.00 and 5.91 (Table 3). No significant differences were observed between limnological parameters and breeding sites analyzed (H = 1.531; $p$ = 0.922).

Fig 5 shows the contribution of eleven limnological variables on the distribution of the three most abundant species collected in the four-water samples. The first axis of the CCA explained between 65.46% and 81.05% of the total variance in species distribution and abundance at breeding sites. The second axis explained between 18.95% and 34.54%. Considering all variables analyzed in the four collections, *A. darlingi* was most affected by pH, total dissolved solids, electrical conductivity, and nitrate. For *A. nuneztovari* s.l., the variables that contributed the most were pH, dissolved oxygen, phosphate, turbidity, and color, while for *A. triannulatus* were temperature and dissolved oxygen.

The negative binomial regression revealed a significant relationship between species abundance and limnological variables. A significant and positive trend was observed between *A. darlingi* and pH (G = 5.685; $\beta$ = 3.733; $p$ = 0.017) and ammonia (G = 5.921; $\beta$ = 6.742; $p$ = 0.014), and between *A. triannulatus* and pH (G = 5.453; $\beta$ = 1.093; $p$ = 0.019), temperature (G = 5.120; $\beta$ = 0.518; $p$ = 0.023), and ammonia (G = 4.957; $\beta$ = 2.491; $p$ = 0.025). For *A. braziliensis*, a significant and negative relationship was observed with nitrate (G = 3.972; $\beta$ = -2.910; $p$ = 0.046), and a positive relationship with turbidity (G = 4.052; $\beta$ = 0.031; $p$ = 0.044) (Table 2).

## Discussion

Of the three types of larval habitats investigated in the District of Ilhâ de Santana, fish farming tanks were the most common and productive for *Anopheles* larvae, with high species richness and abundance. The GLMs indicated that fish farming tanks are the main breeding sites of *A. darlingi* and *A. nuneztovari* s.l., as previously observed for *A. darlingi* at different sites in the Brazilian Amazon [16,33,34]. In the state of Roraima, Barros and Honório [17] found a significantly higher number of larvae in fish farming tanks compared to streams. In malaria-endemic areas in the states of Acre and Amazonas, fish farming tanks had up to four times more anopheline larvae than natural breeding sites [33,35]. In recent years, Saraiva et al. [34], Reis et al. [35], and Maheu-Giroux et al. [36] highlighted the risk of an increase in the number of malaria cases associated with fish farming tanks. Those authors described them as preferred breeding sites for several species of *Anopheles*, especially *A. darlingi*. This contributes to the continuous malaria transmission [15,16,33,34,36].

Barros and Honório [17] estimated that approximately 25% of emerging mosquitoes had a dispersion radius of up to 450 m in Roraima. In Porto Velho, Rondônia, Reis et al. [35] found the highest abundance of *A. darlingi* up to 100 m from human settlements. The total absence of malaria cases was reported at least 900 m distant from fish farming tanks. In the present study, the maximum distance from fish farming tanks to the nearest settlement was 75 m, which is within the radius of mosquito flight dispersion. These conditions favor the anopheline vector feeding on humans and the transmission of malaria in Ilha of Santana.

A significant and negative relationship was observed between species richness and the distance from breeding sites to the nearest human settlement. Thus, breeding sites located more distant from settlements (rural area) had greater species richness. *Anopheles medialis*, *A. mattogrossensis*, and *A. peryassui* were found at these sites, which resulted in greater species richness and confirmed the preference of these species for forest environments. *Anopheles nuneztovari* s.l. and *A. triannulatus*, on the other hand, were collected in great abundance close to settlements. These species are known to adapt well to anthropogenic environments

[37], although they are predominantly zoophilic. Nevertheless, *A. nuneztovari* s.l. was indicated as a malaria vector in Amapá [2,7].

The Kruskal-Wallis's test also revealed how fish farming tanks provide favorable conditions for breeding, especially for *A. darlingi* and *A. nuneztovari* s.l., as they had the following conditions: shading higher than 50%, polluted or murky waters, presence of debris, absence of current, and presence of floating vegetation. In addition to the type of breeding site, this study revealed that shading is an essential variable to maintain larvae anopheline abundance, especially *A. darlingi*. Several studies report that *A. darlingi* larvae prefer habitats with clean water and low sun exposure or shaded [5,37]. In Colombia, Brochero et al. [38] stated that *A. darlingi* larvae were most frequently found at shaded breeding sites, in clean slow flowing water.

However, Barros and Honório [17] stated that the low luminosity may be considered a secondary factor. Those authors considered the absence of water current as the fundamental parameter in the type of ecotone studied. The flowing water in streams, characterized by a turbulent flow with strong currents generated by intense rains, which are more pronounced during the rainy season, can induce larval mortality [17]. Water accumulation, with more stable features throughout the year, was more frequently observed in fish farming tanks, making them more favorable for larval development. Adult female *Anopheles* usually do not lay their eggs in waters with strong currents, as larvae are not adapted to such a condition [21].

We found that the limnological parameters did not indicate favorable conditions for the maintenance of aquatic life in natural water bodies (ponds and streams). Out of nine parameters established by CONAMA, only four were within the range of reference values, indicating that these conditions are unfavorable to natural predators of anopheline or others culicid larvae.

According to CCA analyses, pH was one of the limnological variables that contributed the most to the abundance of *A. darlingi*; ideal pH for maintenance of aquatic life ranges from 6.0 to 9.0 [39].

The GLM-negative binomial regression indicated that *A. darlingi* is best adapted to higher pH values. In general, the pH varied from slightly acidic to neutral, possibly influenced by soil characteristics, presence of organic matter and rainfall [29]. The pH reduction at the breeding sites influences the number of species and the biomass [40]. Regardless of the type of breeding site, pH seems to be a limiting factor, as no anopheline was found at breeding sites with pH below 5.91. A similar result was reported by Barros [41].

Total dissolved solids also affected the abundance of *A. darlingi*. The values obtained for this parameter were all in accordance with CONAMA. This result indicates excellent environmental aquatic conditions, as values above upper limits might be due to recent organic contamination with domestic, industrial or solid waste caused by erosion on the margins of water collections, compromising aquatic life [42]. Another parameter that influenced the abundance of *A. darlingi* was nitrate. This macronutrient is essential for algal growth and is in line with the values established by CONAMA [43]. At low concentrations, nitrate is a limiting factor for the development of aquatic life and high concentrations cause eutrophication [44]. The controlled growth of aquatic plants creates environmental conditions favorable for the development of *Anopheles* larvae.

Similarly, pH, dissolved oxygen, phosphate, turbidity, and color affected *A. nuneztovari* s.l. abundance, although results for the negative binomial regression were not significant. Nevertheless, these parameters influence aquatic life and affect this species. Although *Anopheles* larvae breathe atmospheric oxygen, water oxygen content influences the solubility of inorganic nutrients needed for the survival and development of *Anopheles* larvae [40]. This parameter might be the most important in water quality, as it is essential for the maintenance of aerobic and facultative aquatic organisms involved in carbon, nitrogen, and sulfur metabolism [29].

Water dissolved oxygen content is dependent of temperatures, as the higher the temperature of the environment, the lower the dissolved oxygen content.

Although most analyzed breeding sites had high phosphate levels (95%), which favor the growth of bacteria that are lethal to larvae, our results indicated that this parameter influenced only *A. nuneztovari* s.l., although GLMs did not reveal a statistically significant correlation. Therefore, phosphate was not a limiting factor for the occurrence and abundance of *A. darlingi*, *A. nuneztovari* s.l., and *A. triannulatus*. Phosphate is one of the main macronutrients for aquatic biological processes and indispensable for algal growth, but it can cause eutrophication at high levels [44]. Turbidity was another parameter that influenced *A. nuneztovari* s.l. abundance, but at most breeding sites (95%) the levels were within those established by CONAMA. Unlike turbidity, at most breeding sites (92.25%), color was not within established values. Color is usually associated with the presence of dissolved solids and colloidal particles, derived from the decomposition of organic matter, iron, and manganese, as well as industrial waste and domestic sewage [44]. *Anopheles nuneztovari* s.l. and *A. triannulatus* may be influenced by this variable, as GLMs revealed a positive, but not significant correlation. *Anopheles darlingi*, *A. nuneztovari* s.l., and *A. triannulatus* seemed to be more tolerant to a surplus amount of organic matter in the water.

For *A. triannulatus*, the parameters that contributed the most to its abundance were nitrite, temperature, dissolved oxygen, total dissolved solids, conductivity, turbidity, and color. However, the negative binomial regression indicated a significant and positive correlation only with pH, temperature, and ammonia. This suggests that *A. triannulatus* larvae can occur in aquatic environments with higher temperatures and pH. Water temperature affects the time required for larval development, which may be shorter in warmer waters [21].

In general, *Anopheles* species are sensitive to pollution. In this study, although CCA analyses revealed the influence of conductivity on *A. darlingi* and *A. triannulatus* larvae, GLMs did not show a significant correlation. Again, *A. darlingi* and *A. triannulatus* larvae showed greater tolerance to waters with a higher concentration of pollutants and minerals, since the increase of conductivity releases ions through the decomposition process [29], causing larval mortality.

Anopheline larvae feed mainly on debris and microalgae. In addition to acting as nutritional support of larvae, the algae are important for the oxygenation of aquatic environments [13]. Changes in the phosphorus and nitrogen (ammonia, nitrate and nitrite) content can trigger eutrophication and result in larval mortality. Some of the parameters we analyzed were beyond the value ranges established by CONAMA, but were still favorable to the development of larvae of some *Anopheles* species. Parameters such as pH, ammonia, temperature, and turbidity had a positive influence, while nitrate negatively influenced the occurrence and abundance of species.

Overall, our findings support previous studies [13,16,17,35,45] that fish farming tanks play an important role as permanent breeding sites with high potential for promoting *Anopheles* larval development, especially for *A. darlingi*. This finding is supported by the higher LIMH values (Table 1) observed for this type of breeding site. Similar values were reported for São José de Ribamar, State of Maranhão (0.64), although Barros et al. [45] captured only 75 *Anopheles* larvae in that locality. In this study, *A. darlingi*, *A. nuneztovari* s.l., and *A. triannulatus* were the species that best adapted to fish farming tanks and maintained a high positivity index throughout the study. Despite their high abundance, these species showed a low ABI, when all water collections are considered. However, based on RBI, *A. darlingi*, *A. nuneztovari* s.l. and *A. triannulatus* were detected at almost all breeding sites, with very high values, revealing the wide distribution of these species and their high tolerance in waters with physicochemical quantities beyond the established levels for the maintenance of aquatic life. According to GBI analyses (Table 1), *Anopheles* larvae can develop naturally in low water volumes, if the

environmental conditions are favorable. The volume of water and the size of the breeding area can be important for the colonization of *Anopheles* larvae, as reported by Dantas [40] for other culicids. Inspections carried out in fish farming tanks in the east and west areas of Manaus, Brazil, by Rodrigues et al. [33] assessed the density of larvae based on LIMH. All 141 tanks monitored by those authors were positive for *Anopheles* larvae, but only *A. darlingi* had a positivity index of 75%. The authors emphasized the importance of these studies for the control of *A. darlingi* larvae at artificial breeding sites (fish farming tanks) in the Amazon. Another study performed in Boa Vista also described a high positivity index (85.70%) for *A. darlingi* larvae [17]. Our findings for *A. darlingi*, *A. nuneztovari* s.l., and *A. triannulatus* corroborate both previous studies [17,33].

Considering all the parameters examined, the following entomological indicators were found: 1) fish farming tanks were the most significant larval habitats for the distribution and abundance of the species collected; 2) larval density was highest for *A. darlingi* and *A. nuneztovari* s.l.; 3) the highest positivity index at breeding sites was observed for *A. darlingi*, *A. nuneztovari* s.l., and *A. triannulatus*; 4) the three species were also the most widely distributed and showed greater tolerance range when the physicochemical parameters were beyond those established for the maintenance of aquatic life, which can cause eutrophication; 5) *Anopheles* larvae develop at a low level of water collections.

## Conclusions

Fish farming tanks are potential artificial habitats for *Anopheles* larvae, offering excellent sites for females to lay their eggs, in particular *A. darlingi*. In Brazilian Amazon, these tanks have contributed to the increase of malaria cases throughout the year, mainly in urban and suburban areas due to urbanization advances toward forests. Vector control strategies focused on these habitats are urgently needed. Finally, public policies by local governments that encourage the establishment of fish farming tanks need to be reviewed.

## Acknowledgments

We thank to José C. C. Mendes (Ministério da Saúde do Brazil), Altino M. Rodrigues (Secretaria da Saúde do Estado do Amapá), Miguel O. Brito Filho (Ministério da Saúde do Brazil), and Marcio Alan dos Santos (Secretaria Municipal de Saúde do Estado do Amapá) by their support in the collection of mosquito immatures and water samples. We also thank MSc Claudinaldo Siqueira Ferreira and Augusto Cesar de Souza Moraes by their support in limnological analyses, and residents of Ilha de Santana (Amapá) for their hospitality.

## Author Contributions

**Conceptualization:** Ledayane Mayana Costa Barbosa, Vera Margarete Scarpassa.

**Data curation:** Vera Margarete Scarpassa.

**Formal analysis:** Ledayane Mayana Costa Barbosa.

**Funding acquisition:** Vera Margarete Scarpassa.

**Investigation:** Ledayane Mayana Costa Barbosa, Vera Margarete Scarpassa.

**Methodology:** Ledayane Mayana Costa Barbosa.

**Project administration:** Vera Margarete Scarpassa.

**Resources:** Vera Margarete Scarpassa.

**Software:** Vera Margarete Scarpassa.

**Supervision:** Vera Margarete Scarpassa.

**Validation:** Vera Margarete Scarpassa.

**Visualization:** Vera Margarete Scarpassa.

**Writing – original draft:** Ledayane Mayana Costa Barbosa.

**Writing – review & editing:** Vera Margarete Scarpassa.

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
