## [Decision Letter · Decision Letter 0]

1 Mar 2023

PONE-D-22-27653

Bionomics and population dynamics of Anopheline immatures from an area dominated by fish farming tanks in eastern Amazonian Brazil

PLOS ONE

Dear Dr. Scarpassa,

Thank you for submitting your manuscript to PLOS ONE. After careful consideration, we feel that it has merit but does not fully meet PLOS ONE’s publication criteria as it currently stands. Therefore, we invite you to submit a revised version of the manuscript that addresses the points raised during the review process.

The reviewers have provided their comments for the improvement of the manuscript. They do raise some further points that I recommend are considered in any subsequent revision. These six key points that would need to be addressed can be summarized as:

1. Review the abstract and introduction, clarifying the problem and scientific gap.

2. Clarify the reason for selecting the study area in the methodology section.

3. Clearly explain the environmental parameters that were analyzed and why the authors did not analyze water salinity.

4. Review terminology when referring to insect life cycle stages.

5. Improve the discussion including potential solutions to the problems encountered.

6. Include study limitations.

We look forward to receiving your revised manuscript.

Kind regards,

Delfina Fernandes Hlashwayo, M.Sc.

Academic Editor

PLOS ONE

“NO - Include this sentence at the end of your statement: The funders had no role in study design, data collection and analysis, decision to publish, or preparation of the manuscript.”

4. Thank you for stating the following in the Acknowledgments/ Funding Section of your manuscript:

“This research was funded by MCTI/INPA (grant number: PRJ 12.311) and by PRÓ-EQUIPAMENTO-CAPES.”

“NO - Include this sentence at the end of your statement: The funders had no role in study design, data collection and analysis, decision to publish, or preparation of the manuscript.”

6. We note that Figure 1 in your submission contain [map/satellite] images which may be copyrighted. All PLOS content is published under the Creative Commons Attribution License (CC BY 4.0), which means that the manuscript, images, and Supporting Information files will be freely available online, and any third party is permitted to access, download, copy, distribute, and use these materials in any way, even commercially, with proper attribution. For these reasons, we cannot publish previously copyrighted maps or satellite images created using proprietary data, such as Google software (Google Maps, Street View, and Earth). For more information, see our copyright guidelines: http://journals.plos.org/plosone/s/licenses-and-copyright.

Natural Earth (public domain): http://www.naturalearthdata.com/.

Reviewers' comments:

Reviewer's Responses to Questions

**Comments to the Author**

1. Is the manuscript technically sound, and do the data support the conclusions?

Reviewer #1: Yes

Reviewer #2: Yes

2. Has the statistical analysis been performed appropriately and rigorously? 

Reviewer #1: Yes

Reviewer #2: Yes

3. Have the authors made all data underlying the findings in their manuscript fully available?

Reviewer #1: Yes

Reviewer #2: Yes

4. Is the manuscript presented in an intelligible fashion and written in standard English?

Reviewer #1: Yes

Reviewer #2: Yes

5. Review Comments to the Author

Reviewer #1: Title : Bionomics and population dynamics of Anopheline immatures from an area dominated by fish farming tanks in eastern Amazonian Brazil

Numbe r : PONE-D-22-27653

General comment:

The authors present data that is important for the development of suitable vector control intervention to mitigate malaria transmission in the area, and therefore deserves publication in the journal if several shortcomings issues below are addressed:

1. Environmental parameters

The authors classified 3 breeding sites observed, namely: Fish farming tanks, natural ponds and stream (Figs 2A-C). Unfortunately, the vegetation that are feasible in the picture was not reported. Water vegetation is also important for some anopheles to avoid direct sun light in the tropical environment.

2. Limnological parameter:

In the Table 3, water salinity was not recorded. Any specific reason why salinity is not measured? Is the study site located far inland?

3. Entomological parameters:

The “immature” stage is better replaced with larva(e).

4. Results of this study underlines the importance of implementation of larval source management (LSM). As the majority of the breeding sites are fish farming tnks, the authors should discuss several options that might be suitable for the site and is acceptable by the owners of fish tank.

Reviewer #2: This paper is well written

Though I have this few comments

REVIEWER COMMENTS

#1 Comment: row number 1

Rewrite abstracts should show problem and aim, methodology, result and conclusion. Write it clearly for a reader to understand the problem and the gap that you are trying to find the solution

#2 comment: row number 1 introduction

When you say highest risk, you should show the number of incidences, and show how it lower the number of malaria cases, which interventions were used to lower interventions. Show the decrease in numbers.

#3 Comment: row number 1 material and methods

You need to explain why you selected the study area. What were the reason to select that area, was it because of higher malaria prevalence, was it because of agricultural activities that influence the presence of immature?

#4 Comments:

What were the limitations of this study? Need to explain the limitations so as to create the ground for those who will conduct this kind of study to take into consideration.

6. PLOS authors have the option to publish the peer review history of their article (what does this mean?). If published, this will include your full peer review and any attached files.

Reviewer #1: No

Reviewer #2: No

---

## [Author Response · Author response to Decision Letter 0]

19 May 2023

Manaus, April 5, 2023.

REBUTTAL LETTER

List of specific response of each Reviewer

Journal: PLoS ONE 

Manuscript Number: PONE-D-22-27653

Title: Bionomics and population dynamics of anopheline larvae from an area dominated by fish farming tanks in northern Amazonian Brazil

Authors: Ledayane Mayana Costa Barbosa; Vera Margarete Scarpassa

Dr. Delfina Fernandes Hlashwayo 

Academic Editor of PLOS ONE 

Dear Dr. Hlashwayo,

We are very grateful to the you and two Reviewers for the comments and observations who significantly improved this manuscript. We hope that this revised version will be acceptable for publication in this important Journal.

We inform that the manuscript was extensively revised encompassing all sections, especially abstract. The title was also slightly modified as well as the English was improved throughout the manuscript, as are listed in “Revised manuscript with track copy”.

We also inform that Satellite image (Figure 1) was obtained from site https://eros.usgs.gov/# USGS EROS (Earth Resources Observatory and Science (EROS) Center) (public dominion) provided by you in the message “PLOS ONE Decision: Revision required [PONE-D-22-27653]”.

The references were revised in accordance the PLoS ONE’s Rules, as are listed in “Revised manuscript with track copy”. 

We provided below a list of specific response for each Reviewer comment.

LIST OF SPECIFIC RESPONSES TO REVIEWER NUMBER 1:

Reviewer’ comments #1: 

Environmental parameters. The authors classified 3 breeding sites observed, namely: Fish farming tanks, natural ponds and stream (Figs 2A-C). Unfortunately, the vegetation that are feasible in the picture was not reported. Water vegetation is also important for some anopheles to avoid direct sun light in the tropical environment.

Author response: Sorry, we forget of include this information into first version of this manuscript. We provided this information “The aquatic vegetation found in breeding sites was identified as Lemma sp., Pistia sp., Eichhornia sp., Nymphaea sp. and Juncus sp. “This information is on page 12, lines 11 and 12.

Reviewer’comments #2: 

Limnological parameter: In the Table 3, water salinity was not recorded. Any specific reason why salinity is not measured? 

Author response: 

We did not calculate the water salinity because the previous Limnological studies have reported that in the Amazon Delta and also for several tens of kilometers, along the coast, in north direction, salinity is close to zero. For this reason, we did not assess salinity in this study.

Reviewer’comments #2: Is the study site located far inland? 

Author response: It is about 817 meters from Inland.

Reviewer’comments #3: 

Entomological parameters: The “immature” stage is better replaced with larva(e).

Author response: We have attended this suggestion. However, there are few sentences into manuscript that we kept the word immature, because encompasses both larval and pupal stages. The pupae were also collected in this study and they were kept in plastic bottles with water from the breeding site until the emergence of adults and following were identified.

Reviewer’comments #4:

4. Results of this study underlines the importance of implementation of larval source management (LSM). As the majority of the breeding sites are fish farming tanks, the authors should discuss several options that might be suitable for the site and is acceptable by the owners of fish tank.

Author response: The ideal situation would be to use the fish farming tanks for the purpose for which they were built. However, the residents do not have the resources or funding from the government to carry out or to maintain this project. Then, the alternative situation and the most indicate and drastic, would be to neutralize or remove these tanks. This measure requires extensive discussion together with the Staffs of the Secretary of Health of the states and communities local. This is a very polemic situation. For this reason, we have limited this discussion into manuscript.

LIST OF SPECIFIC RESPONSES TO REVIEWER NUMBER 2:

Reviewer’general comment

This paper is well written. Though I have this few comments.

Author response:

Thank you

Reviewer’comments #1:

Rewrite abstracts should show problem and aim, methodology, result and conclusion. Write it clearly for a reader to understand the problem and the gap that you are trying to find the solution

Author response:

We have included these topics into abstract. Please, see page 2.

Reviewer’comments #2:

Introduction

When you say highest risk, you should show the number of incidences, and show how it lower the number of malaria cases, which interventions were used to lower interventions. Show the decrease in numbers.

Author response: We have attended this suggestion. These informations were included in first paragraph from Introduction section, page 3.

Reviewer’comments #3:

row number 1 material and methods You need to explain why you selected the study area. What were the reasons to select that area, was it because of higher malaria prevalence, was it because of agricultural activities that influence the presence of immature?

Author response: We have attended this suggestion. The sentence was included on page 5, lines 1, 2 and 3 from M&M section, as follow: “The selection of the study area took into account number of malaria cases, high incidence of A. darlingi and others vector [2, 7-9,18] and the high migratory flow of humans related to intense agricultural activities.”

Reviewer’comments #4:

What were the limitations of this study? Need to explain the limitations so as to create the ground for those who will conduct this kind of study to take into consideration.

Author response: The difficulties found are common to other studies, such as limited or scarce funds (grants) and lack of field staff for help us. For this study, we had to pay a team to help us in field with our own salary. It was too hardy.

---

## [Editor Report · Decision Letter 1]

23 May 2023

PONE-D-22-27653R1Bionomics and population dynamics of anopheline larvae from an area dominated by fish farming tanks in northern Amazonian BrazilPLOS ONE

Dear Dr. Scarpassa,

Thank you for submitting your manuscript to PLOS ONE. After careful consideration, we feel that it has merit but does not fully meet PLOS ONE’s publication criteria as it currently stands. Therefore, we invite you to submit a revised version of the manuscript that addresses the points raised during the review process.

We appreciate the effort you have made in addressing the comments provided by the reviewers. We have found the revisions to be satisfactory. Before proceeding with the final decision, we would like to suggest that you consider conducting a proofreading of the manuscript. Proofreading plays a crucial role in ensuring the accuracy and clarity of the content. It allows you to review the grammar, spelling, and overall language usage, ensuring that the article is presented in the best possible form. A thorough proofreading can further enhance the quality and readability of your work. It will help eliminate any potential minor errors or inconsistencies that may have been overlooked during the initial review process. 

We appreciate your understanding and cooperation in this matter. 

We look forward to receiving your revised manuscript.

Kind regards,

Delfina Fernandes Hlashwayo, M.Sc.

Academic Editor

PLOS ONE
---

## [Author Response · Author response to Decision Letter 1]

4 Jul 2023

Manaus, April 5, 2023.

REBUTTAL LETTER

List of specific response of each Reviewer

Journal: PLoS ONE 

Manuscript Number: PONE-D-22-27653

Title: Bionomics and population dynamics of anopheline larvae from an area dominated by fish farming tanks in northern Amazonian Brazil

Authors: Ledayane Mayana Costa Barbosa; Vera Margarete Scarpassa

Dr. Delfina Fernandes Hlashwayo 

Academic Editor of PLOS ONE 

Dear Dr. Hlashwayo,

We are very grateful to the you and two Reviewers for the comments and observations who significantly improved this manuscript. We hope that this revised version will be acceptable for publication in this important Journal.

We inform that the manuscript was extensively revised encompassing all sections, especially abstract. The title was also slightly modified as well as the English was improved throughout the manuscript, as are listed in “Revised manuscript with track copy”.

We also inform that Satellite image (Figure 1) was obtained from site https://eros.usgs.gov/# USGS EROS (Earth Resources Observatory and Science (EROS) Center) (public dominion) provided by you in the message “PLOS ONE Decision: Revision required [PONE-D-22-27653]”.

The references were revised in accordance the PLoS ONE’s Rules, as are listed in “Revised manuscript with track copy”. 

We provided below a list of specific response for each Reviewer comment.

LIST OF SPECIFIC RESPONSES TO REVIEWER NUMBER 1:

Reviewer’ comments #1: 

Environmental parameters. The authors classified 3 breeding sites observed, namely: Fish farming tanks, natural ponds and stream (Figs 2A-C). Unfortunately, the vegetation that are feasible in the picture was not reported. Water vegetation is also important for some anopheles to avoid direct sun light in the tropical environment.

Author response: Sorry, we forget of include this information into first version of this manuscript. We provided this information “The aquatic vegetation found in breeding sites was identified as Lemma sp., Pistia sp., Eichhornia sp., Nymphaea sp. and Juncus sp. “This information is on page 12, lines 11 and 12.

Reviewer’comments #2: 

Limnological parameter: In the Table 3, water salinity was not recorded. Any specific reason why salinity is not measured? 

Author response: 

We did not calculate the water salinity because the previous Limnological studies have reported that in the Amazon Delta and also for several tens of kilometers, along the coast, in north direction, salinity is close to zero. For this reason, we did not assess salinity in this study.

Reviewer’comments #2: Is the study site located far inland? 

Author response: It is about 817 meters from Inland.

Reviewer’comments #3: 

Entomological parameters: The “immature” stage is better replaced with larva(e).

Author response: We have attended this suggestion. However, there are few sentences into manuscript that we kept the word immature, because encompasses both larval and pupal stages. The pupae were also collected in this study and they were kept in plastic bottles with water from the breeding site until the emergence of adults and following were identified.

Reviewer’comments #4:

4. Results of this study underlines the importance of implementation of larval source management (LSM). As the majority of the breeding sites are fish farming tanks, the authors should discuss several options that might be suitable for the site and is acceptable by the owners of fish tank.

Author response: The ideal situation would be to use the fish farming tanks for the purpose for which they were built. However, the residents do not have the resources or funding from the government to carry out or to maintain this project. Then, the alternative situation and the most indicate and drastic, would be to neutralize or remove these tanks. This measure requires extensive discussion together with the Staffs of the Secretary of Health of the states and communities local. This is a very polemic situation. For this reason, we have limited this discussion into manuscript.

LIST OF SPECIFIC RESPONSES TO REVIEWER NUMBER 2:

Reviewer’general comment

This paper is well written. Though I have this few comments.

Author response:

Thank you

Reviewer’comments #1:

Rewrite abstracts should show problem and aim, methodology, result and conclusion. Write it clearly for a reader to understand the problem and the gap that you are trying to find the solution

Author response:

We have included these topics into abstract. Please, see page 2.

Reviewer’comments #2:

Introduction

When you say highest risk, you should show the number of incidences, and show how it lower the number of malaria cases, which interventions were used to lower interventions. Show the decrease in numbers.

Author response: We have attended this suggestion. These informations were included in first paragraph from Introduction section, page 3.

Reviewer’comments #3:

row number 1 material and methods You need to explain why you selected the study area. What were the reasons to select that area, was it because of higher malaria prevalence, was it because of agricultural activities that influence the presence of immature?

Author response: We have attended this suggestion. The sentence was included on page 5, lines 1, 2 and 3 from M&M section, as follow: “The selection of the study area took into account number of malaria cases, high incidence of A. darlingi and others vector [2, 7-9,18] and the high migratory flow of humans related to intense agricultural activities.”

Reviewer’comments #4:

What were the limitations of this study? Need to explain the limitations so as to create the ground for those who will conduct this kind of study to take into consideration.

Author response: The difficulties found are common to other studies, such as limited or scarce funds (grants) and lack of field staff for help us. For this study, we had to pay a team to help us in field with our own salary. It was too hardy.

---

## [Editor Report · Decision Letter 2]

10 Jul 2023

Bionomics and population dynamics of anopheline larvae from an area dominated by fish farming tanks in northern Brazilian Amazon

PONE-D-22-27653R2

Dear Dr. Scarpassa,

We’re pleased to inform you that your manuscript has been judged scientifically suitable for publication and will be formally accepted for publication once it meets all outstanding technical requirements.

Kind regards,

Delfina Fernandes Hlashwayo, M.Sc.

Academic Editor

PLOS ONE

Additional Editor Comments (optional):

The tables present in the main text are identical to those in the supplemental files. Therefore, I recommend considering keeping them in either the main text or the supplemental files to avoid redundancy. Additionally, the figure included in the supplemental files is also present in the main text. I kindly ask you to decide whether it should remain in the supplemental files or be included solely in the main text, based on the most appropriate location for a comprehensive understanding of the study.
---

## [Editor Report · Acceptance letter]

17 Aug 2023

PONE-D-22-27653R2 

Bionomics and population dynamics of anopheline larvae from an area dominated by fish farming tanks in northern Brazilian Amazon 

Dear Dr. Scarpassa:

I'm pleased to inform you that your manuscript has been deemed suitable for publication in PLOS ONE. Congratulations! Your manuscript is now with our production department. 

Kind regards, 

on behalf of

Ms. Delfina Fernandes Hlashwayo 

Academic Editor

PLOS ONE